

# Decoherence and pointer states in small antiferromagnets: A benchmark test

Hylke C. Donker[1*], Hans De Raedt[2] and Mikhail I. Katsnelson[1]

**1** Radboud University, Institute for Molecules and Materials, Heyendaalseweg 135, NL-6525AJ Nijmegen, the Netherlands
**2** Zernike Institute for Advanced Materials, University of Groningen, Nijenborgh 4, NL-9747AG Groningen, the Netherlands

\* h.donker@science.ru.nl

## Abstract

We study the decoherence process of a four spin-1/2 antiferromagnet that is coupled to an environment of spin-1/2 particles. The preferred basis of the antiferromagnet is discussed in two limiting cases and we identify two *exact* pointer states. Decoherence near the two limits is examined whereby entropy is used to quantify the *robustness* of states against environmental coupling. We find that close to the quantum measurement limit, the self-Hamiltonian of the system of interest can become dynamically relevant on macroscopic timescales. We illustrate this point by explicitly constructing a state that is more robust than (generic) states diagonal in the system-environment interaction Hamiltonian.

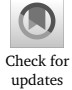

# 1 Introduction

Understanding the interplay between a quantum system and its environment is of utmost importance both from an engineering and a fundamental point of view. On the one hand, large coherence times are required to engineer quantum computers in which system-environment coupling is largely an undesired effect [1]. On the other hand, environmental coupling opens up the possibility to gain insight in the quantum-to-classical transition in which decoherence is thought to be the key ingredient [2, 3].

In either case it is of interest to understand which states of the system under study – henceforth central-system (CS) – are least prone to environmental deterioration. In general, it is believed that these environmentally robust states, so-called pointer states (PS), emerge from the interplay between the *de*cohering effect of the interaction with the environment and the *re*cohering dynamics of the CS [3]. Progress has been made to identify particular PS corresponding to two opposite limits [4]:

1. In the strong-coupling case, the dynamical time-scale of the CS Hamiltonian $H_S$ is assumed to be completely negligible compared to the interaction Hamiltonian $H_I$. In this case the PS are determined by the basis of the interaction Hamiltonian $H_I$.

2. In the opposite limit, the interaction Hamiltonian $H_I$ is taken to be small and (adiabatically) slowly varying compared to the CS, i.e. the dynamics are dominated by $H_S$. In this case the PS are found to coincide with the energy eigenstates of $H_S$, regardless of the specific form of the interaction.

These two limits, referred to as limit 1 and 2 throughout, were analysed in Refs. [2, 3, 5–10] and [4, 5, 8–16], respectively. However, studies that emphasise entropy as a criterion for PS (to be discussed in more detail in the next section) [17] have been primarily restricted to a few exactly solvable master equations [3, 11, 18, 19]. In addition, new experiments demonstrate that concepts such as purity and entropy have become observable quantities [20, 21].

In this work, the decoherence process is examined by numerically evaluating the Schrödinger time-evolution of a collection of spin-1/2 particles. Motivated by recent progress in spin polarised STM experiments [22] (see [23] for a recent review), the present work shall encompass the decoherence of a small antiferromagnetic CS. Importantly, not only the loss of coherence, but also the production of entropy is stressed. The two aforementioned opposite regimes are considered by examining decoherence close to the respective idealised limits.

This paper is organised as follows: The characteristics of PS are briefly reviewed in Sec. 2, and an antiferromagnetic model is introduced in Sec. 3. Using this model, the concept of PS are illustrated in Sec. 4 by working out explicitly an idealised decoherence process. The main numerical results are presented and interpreted in Sec. 5. These results are subsequently

discussed in Sec. 6 with particular emphasis on the relation to classicality. And finally the main findings are recapitulated in the Conclusion.

## 2 Pointer states

The reduced density matrix (RDM) associated with the CS is usually of primary interest when studying decoherence in general and PS in specific. This RDM is obtained from the (pure) density matrix of the entire system (i.e., both the CS and the environment) by tracing out environmental degrees of freedom

$$\rho(t) = \mathrm{Tr}_{\mathscr{E}}\Pi(t), \tag{1}$$

where $\Pi(t) = |\Psi(t)\rangle\langle\Psi(t)|$ denotes the density matrix of the combined system. In order to find the pointer basis (assuming that such a basis exists at all), it does not suffice to perform straightforward diagonalisation of the RDM $\rho(t)$. Any RDM can trivially be brought to diagonal form by using an appropriate unitary transformation. As was emphasised by Zurek [17], diagonality of the RDM is only a symptom of preferred states. And indeed, the (instantaneous) Schmidt basis of a RDM can be radically different from the pointer basis, as was stressed by Schlosshauer [24].

Instead, the decoherence program has put forward the idea that a preferred basis of a system is singled out by the environment [2, 3]. The preferred states are selected based on the requirement that correlations are best preserved [2, 17, 24, 25]. Originally, the concept of the pointer basis was discussed in the setting of the system-apparatus-environment triad [25], which will now serve as an useful example.

Let $\{|s_n\rangle\}$, $\{|A_n\rangle\}$, and $\{|\mathscr{E}_n\rangle\}$ refer to basis vectors of the system, apparatus, and environment, respectively. In the ideal case in which system-apparatus correlations can be maintained, the coupling to the environment would result, for example, in the development of entanglement of the form

$$\left(\sum_n c_n|s_n\rangle|A_n\rangle\right)|\mathscr{E}_0\rangle \longrightarrow \sum_n c_n|s_n\rangle|A_n\rangle|\mathscr{E}_n\rangle, \tag{2}$$

with $|\mathscr{E}_0\rangle$ the initial state of the environment. The formation of such correlations can typically be achieved by considering e.g. a von Neumann-type of system-environment interaction [26]. More often than not, however, the development is not as ideal as schematically depicted in Eq. (2). And generally, the states $|s_n\rangle$ are perturbed and evolve into different states $|\tilde{s}_n(t)\rangle$ (N.B. we consider the effect of the environment onto the states $\{|s_n\rangle\}$). Consequently, the initial $|s_n\rangle|A_n\rangle$ correlation diminishes over time.

From this example it becomes clear that states must be *robust* against environmental interactions, in order to preserve correlations. Thus, the pointer basis consists of states which are least affected by the environment. The robustness of preferred states is perhaps made more concrete in Schrödinger's cat paradox. In this gedankenexperiment the preservation of correlations means that a dead cat must remain dead, even when one decides to illuminate the dead cat to have a look and ascertain its physiological state; that is, environmental photons scattering off the dead cat towards our eyes should leave the state of the cat essentially unperturbed.

To understand why correlation preservation is assumed to be a characteristic of classicality, it is useful to think of the coupling to the environment as if it is a measurement (more detailed analysis [27, 28], however, indicates that more intricate conditions are required to genuinely speak of measurement). One of the characteristic features of classical physics is that any disturbances resulting from a measurement can, at least in principle, be made arbitrarily small [29]. This is often taken as one of the defining features [3, 17] of effective classicality: would-be

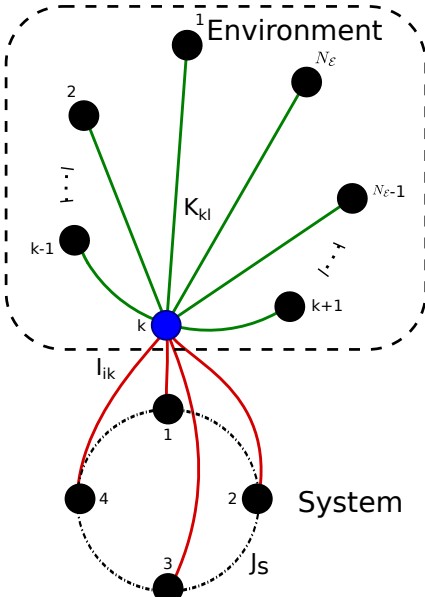

Figure 1: Schematic set-up of the spin-spin interactions as described by the Hamiltonian $H$, Eq. (4). A single representative environment particle, $k$, is highlighted (in blue) to illustrate the intra-environment coupling (green solid lines) and the coupling to the central-system (red solid lines). Each solid line denotes an antiferromagnetic interaction of random strength. All other environment particles have analogous coupling as illustrated by particle $k$. The dot-dashed lines in the central-system represent the nearest-neighbour coupling in $H_S$ of (constant) strength $J_S$.

classical states are insensitive to measurement of classical observables. In this sense, states which are robust against the *measuring effect* of the environment, are effectively classical.

One of the methods that was proposed to identify PS is the predictability sieve criterion [2,17] (for alternative criteria see e.g. Refs. [18,19]). For completeness we briefly outline this algorithmic procedure, closely following [2,17]. The first step is to make a list of all possible pure states in the relevant Hilbert space. One subsequently evaluates the von Neumann entropy [26]

$$S(t) \equiv -\text{Tr}\left[\rho(t)\ln\rho(t)\right], \tag{3}$$

for each element in the list by letting the system interact with the environment for some fixed value of $t$. Finally, the list is to be ordered in descending value of $S(t)$ and one requires that the states on the top of the list (with lowest $S(t)$) do not change appreciably for variations in $t$. As a result, the top of the list contains the states that are the least susceptible to environmental deterioration and therefore outperform other states in terms of retaining correlations.

In practice, it is rather difficult to numerically evaluate the entropy production of each possible linear combination. Therefore, we restrict our analysis to specific states where predictions, as to whether this state is preferred or not, are available.

## 3 Model

### 3.1 Hamiltonian

In this work we will study an ensemble of magnetic particles which are modelled as spin-1/2 states. The ensemble is partitioned in two: the system of interest (the CS) and its environment.

We therefore write the Hamiltonian of the entire system as a sum of three parts:

$$H = H_S + H_I + H_{\mathscr{E}}, \tag{4}$$

whereby $H_S$ ($H_{\mathscr{E}}$) refers to the Hamiltonian of the CS (environment) and $H_I$ contains the system-environment coupling. The CS consists of $N_S$ antiferromagnetically coupled spins, governed by the Heisenberg Hamiltonian [30]

$$H_S = J_S \sum_{\langle i,j \rangle \in S} \boldsymbol{S}_i \cdot \boldsymbol{S}_j, \tag{5}$$

where $J_S$ is the exchange integral ($J_S > 0$ for antiferromagnetism), $i$ and $j$ are indices in the subset S pertaining to the CS, and $\langle i,j \rangle$ indicates that the two indices refer to nearest neighbours. Henceforth units in which $\hbar = 1$ are used so that the spin operators are given by $S_i^{\alpha} = \sigma_i^{\alpha}/2$, with $\sigma^{\alpha}$ Pauli matrices. Similarly, the interaction Hamiltonian is chosen to be

$$H_I = \sum_{i \in S, k \in \mathscr{E}} I_{ik} S_i^z S_k^z, \tag{6}$$

with $I_{ik}$ the coupling strength between spins $i$ and $k$, where $i$ is in the subset $S$ and $k$ belongs to the subset of (in total $N_{\mathscr{E}}$) environment indices $\mathscr{E}$. As a consequence of the Ising-coupling in Eq. (6), the total $z$-magnetization, $S_{\text{tot}}^z = \sum_{i \in S} S_i^z$, of the CS is conserved: $[S_{\text{tot}}^z, H] = 0$ [see Eq. (4)].

From experimental decoherence studies it is known that system-environment couplings are, in certain cases, well captured using random strengths [31–33]. In addition, numerical calculations indicate that the use of random couplings enhance the decoherence process [34]. This motivates us to model the coupling between the CS and the environment with a random interaction

$$I_{ab} = I r_{ab}, \tag{7}$$

where $I$ denotes the strength of the interaction and $r_{ab}$ are uniform random numbers in the range $[0,1)$ such that each $r_{ab}$ is a different realisation for each $a$ and $b$.

The Hamiltonian of the environment is set to

$$H_{\mathscr{E}} = \sum_{k,l \in \mathscr{E}} K_{kl} \boldsymbol{S}_k \cdot \boldsymbol{S}_l, \tag{8}$$

where $K_{kl}$ denotes the intra-environment interaction strength. Similar as for $H_I$, the use of random intra-environment strengths [34] and large connectivity [14] allows one to achieve optimal loss of coherence. We corroborate the findings of [14,34] from extensive simulations, the results of which are not shown here. Incorporating these features, the interaction strengths $K_{kl}$ of $H_{\mathscr{E}}$ [Eq. (8)] takes the form

$$K_{ab} = K \tilde{r}_{ab}. \tag{9}$$

Here $\tilde{r}_{ab}$ denotes uniform random numbers in the range $[0,1)$ analogous to $r_{ab}$. To schematically depict Eq. (4), we show the system-environment and environment-environment connections of a single representative environment particle $k$ in Fig. 1.

## 3.2 State preparation

Since the process of decoherence arises from the development of entanglement, we find it convenient to prepare the entire system in a product state at time $t = 0$:

$$\Pi(t = 0) = \rho_S \otimes \rho_{\mathscr{E}}, \tag{10}$$

with $\rho_S$ the density matrix of the CS and $\rho_{\mathscr{E}}$ that of the environment. Subsequent time-evolution is governed by the unitary operator $U(t) = \exp[-iHt]$ as described by the Schrödinger equation.

Inspired by experiments [31,32], the environment, $\rho_{\mathscr{E}} = |\phi\rangle\langle\phi|$, is initially prepared in a state corresponding to temperature $T = \infty$. This is done by constructing the (normalised) state $|\phi\rangle = \sum_{n=1}^{2^{N_{\mathscr{E}}}} z_n |n\rangle$, where $z_n$ is a random complex number such that $|\mathbf{z}|^2 = 1$ (see Sec. 3.3).

As for the initial state of the CS, our attention shall be restricted to specific initial configurations. Two states will be studied in particular: the Néel state $|\psi_N\rangle = |\uparrow\downarrow\uparrow\ldots\rangle$ (the arrows denote spin-up and -down in the computational- or Ising basis) and the ground state of $H_S$ called $|\psi_0\rangle$. These two initial states are particularly convenient for scrutinizing PS, since they coincide with the preferred basis in limits 1 and 2, respectively. Moreover, the former corresponds to the classical limit of the latter, as will be discussed in more detail in Sec. 6.

However, a direct consequence of choosing initial states $|\psi_0\rangle$ and $|\psi_N\rangle$, whereby both states belong to the $S_{\text{tot}}^z = 0$ subspace, is that a $N_S = 2$ particle CS can effectively be described by a two-level system. In this case the entropy directly follows from a single eigenvalue of $\rho(t)$ (the other eigenvalue is fixed by normalisation). We shall therefore focus on a slightly larger $N_S = 4$ CS for which the entropy is less constrained.

### 3.3 Simulation procedure

The unitary time evolution of the wave function $|\Psi(t)\rangle$, or equivalently the evolution of the density matrix $\Pi(t) = |\Psi(t)\rangle\langle\Psi(t)|$, is calculated using the Chebyshev polynomial expansion, which yields numerically exact results up to machine precision [35,36]. The Box-Muller method [37] is used to generate the random coefficients $z_n$ for the initial state of the environment. The choice of the Heisenberg Hamiltonian for the CS allows for certain sanity checks. For example, commuting of $S_{\text{tot}}^z$ with the Hamiltonian means that the total magnetization is conserved, as already discussed. Moreover, one can use (e.g.) the exact PS to verify that they remain unaltered upon time-evolution (see Sec. 4).

The simulations presented in Sec. 5 are carried out using a single realisation of random couplings [Eqs. (7) and (9)]. Simulations in which $N_{\mathscr{E}}$ is varied (Secs. A and B) use different realisations of the couplings. For each run (corresponding to inequivalent $\{N_S, N_{\mathscr{E}}, \rho(t=0), I, K\}$) a new environment state $|\phi\rangle$ is generated. The phenomena observed in this work are insensitive to different realisations of the couplings as well as the state of the environment $|\phi\rangle$.

## 4 Example: exact pointer states

To illustrate the characteristic features of PS, it is illuminating to start with an essentially trivial example: *exact* pointer states. With *exact* we indicate that these states are simultaneously eigenstates of the CS, $H_S$, and the interaction Hamiltonian, $H_I$. Two important aspects can now be emphasised: the loss of coherence and the production of entropy. Hamiltonian $H$ [Eq. (4)] allows one to identify at least two exact PS: namely, the fully polarised states $|\Uparrow\rangle = |\uparrow\uparrow\uparrow\ldots\rangle$ and $|\Downarrow\rangle = |\downarrow\downarrow\downarrow\ldots\rangle$. Such a state is stationary and produces no entropy, as we will show now.

Take, as an example, the initial state $|\Psi(0)\rangle = |\Uparrow\rangle|\psi_{\mathscr{E}}\rangle$, whereby $|\psi_{\mathscr{E}}\rangle$ is an arbitrary initial state of the environment. Evolution of $\Pi(t) = |\Psi\rangle\langle\Psi|$ is governed by the von Neumann equation and thus the time-dependence of $\rho(t)$ is determined by

$$\frac{\hbar}{i}\text{Tr}_{\mathscr{E}}\left[\frac{\partial\Pi}{\partial t}\right] = \text{Tr}_{\mathscr{E}}\left[\Pi, H_S + H_I + H_{\mathscr{E}}\right]. \tag{11}$$

It follows that the commutator evaluates to zero, as is seen from the cyclic property of trace and the fact that the initial state is an eigenstate of both $H_S$ and $H_I$. Hence, this state retains

its purity and is resilient to environmental coupling, irrespective of the numerical values of $I_{ab}$ and $K_{ab}$. Conversely, for more general superpositions of $|\Uparrow\rangle$ and $|\Downarrow\rangle$ entanglement will develop with the environment. The production of entanglement generates entropy and suppresses coherences in $S$ as a result of tracing over environmental degrees of freedom. In order to show the formation of entanglement, first note that:

$$e^{-it(H_S+H_I+H_{\mathscr{E}})}|\Uparrow\rangle|\psi_{\mathscr{E}}\rangle = |\Uparrow\rangle\left(\sum_{n=0}^{\infty}\frac{\left(-it\left[E_S^{\Uparrow}+\sum_{k\in\mathscr{E}}\alpha_k S_k^z+H_{\mathscr{E}}\right]\right)^n}{n!}\right)|\psi_{\mathscr{E}}\rangle$$

$$= |\Uparrow\rangle e^{-it\left(E_S^{\Uparrow}+\sum_{k\in\mathscr{E}}\alpha_k S_k^z+H_{\mathscr{E}}\right)}|\psi_{\mathscr{E}}\rangle\,, \tag{12}$$

where $E_S^{\Uparrow} = E_S^{\Downarrow} = J_S N_S/4$ and the symbol $\alpha_k = \sum_{l\in S}I_{lk}/2$ was introduced to denote an eigenvalue of the CS state $|\Uparrow\rangle$. Using this identity one can evaluate the time-evolution of a more general linear combination $|\Upsilon(0)\rangle = (\alpha|\Uparrow\rangle + \beta|\Downarrow\rangle)|\psi_{\mathscr{E}}\rangle$. Following the notation in Refs. [2,8] one finds

$$|\Upsilon(t)\rangle = e^{-it(H_S+H_I+H_{\mathscr{E}})}(\alpha|\Uparrow\rangle|\psi_{\mathscr{E}}\rangle + \beta|\Downarrow\rangle|\psi_{\mathscr{E}}\rangle) \tag{13}$$

$$= e^{-itE_S^{\Uparrow}}\left(\alpha|\Uparrow\rangle|\mathscr{E}_{\Uparrow}(t)\rangle + \beta|\Downarrow\rangle|\mathscr{E}_{\Downarrow}(t)\rangle\right)\,, \tag{14}$$

where

$$|\mathscr{E}_{\Uparrow}(t)\rangle = \exp\left[-it\left(\sum_{k\in\mathscr{E}}+\alpha_k S_k^z+H_{\mathscr{E}}\right)\right]|\psi_{\mathscr{E}}\rangle\,, \tag{15}$$

$$|\mathscr{E}_{\Downarrow}(t)\rangle = \exp\left[-it\left(\sum_{k\in\mathscr{E}}-\alpha_k S_k^z+H_{\mathscr{E}}\right)\right]|\psi_{\mathscr{E}}\rangle\,. \tag{16}$$

The transition of $\rho(t)$ from a pure to a mixed state becomes explicit by tracing out the environment

$$\rho(t) = \text{Tr}_{\mathscr{E}}[|\Upsilon(t)\rangle\langle\Upsilon(t)|] = \begin{pmatrix} |\alpha|^2 & \langle\mathscr{E}_{\Downarrow}(t)|\mathscr{E}_{\Uparrow}(t)\rangle\alpha\beta^* \\ \langle\mathscr{E}_{\Uparrow}(t)|\mathscr{E}_{\Downarrow}(t)\rangle\alpha^*\beta & |\beta|^2 \end{pmatrix}. \tag{17}$$

Decoherence is successful if $\langle\mathscr{E}_{\Uparrow}(t)|\mathscr{E}_{\Downarrow}(t)\rangle \approx 0$ which results in the production of an entropy $\Delta S = -[p\ln p + (1-p)\ln(1-p)]$, where $p = |\alpha|^2$.

## 5 Results

One of the quantities that shall be used to characterise PS is entropy $S(t)$, as discussed in Sec. 2. In addition, we introduce the basis-dependent symbol $\mathscr{M}(t)$ to denote, for each time step $t$ and all $i$ and $j$, the maximum off-diagonal component $|\rho_{i\neq j}|$,

$$\mathscr{M}(t) \equiv \max\left[|\rho_{i\neq j}(t)|\right]\,, \tag{18}$$

whereby the indices $i$ and $j$ refer to a particular basis. Although the quantity $\mathscr{M}(t)$ serves as a convenient gauge for the development of coherence in a particular basis, different measures such as the expectation value, $E[|\rho_{i\neq j}(t)|]$, or the standard deviation, $\sigma[|\rho_{i\neq j}(t)|]$, were found to be equally good measures for (the lack off) coherence. Both $S(t)$ and $\mathscr{M}(t)$ shall now be used as the two guiding quantities in evaluating the pointer state-like behaviour of specific states.

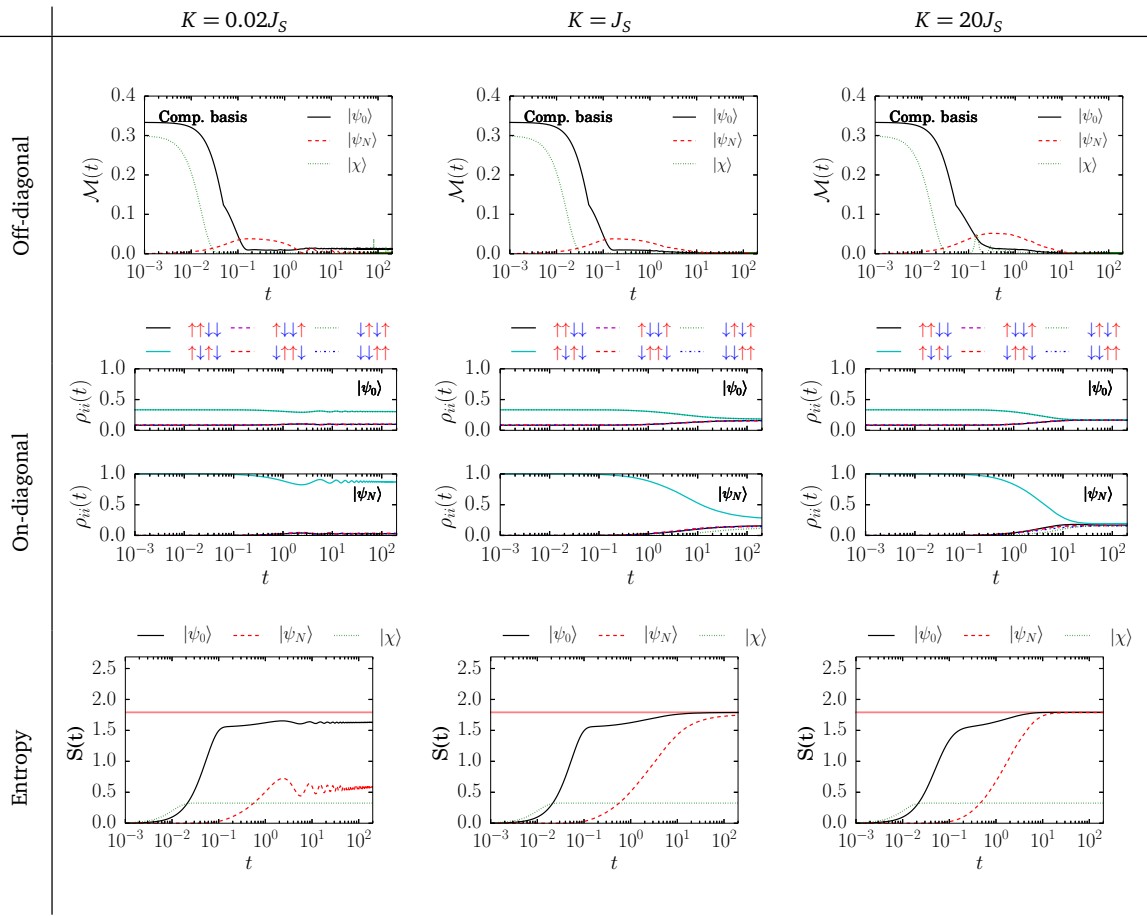

Figure 2: Time-evolution of the reduced density matrix $\rho_{ij}(t)$, expressed in the computational basis, for different intra-environment strengths, $K$, (columns) and initial states of the central-system (to wit, ground state $|\psi_0\rangle$, Néel-state $|\psi_N\rangle = |\uparrow\downarrow\uparrow\downarrow\rangle$, and $|\chi\rangle = [|\Uparrow\rangle + 3|\Downarrow\rangle]/\sqrt{10}$). The central-system consists of $N_S = 4$ spins that are connected to each of the $N_{\mathscr{E}} = 16$ environment spins via random antiferromagnetic Ising coupling of strength $I = 20J_S$. The top row indicates the maximum off-diagonal components $\mathscr{M}(t)$ [Eq. (18)] as function of dimensionless time $t$. The middle row depicts the diagonal components of $\rho(t)$ in the $S^z_{\text{tot}} = 0$ subspace, as indicated by the spin configurations in the legend. The von Neumann entropy of $\rho(t)$ is shown in the bottom row, and the red horizontal line indicates the maximal entropy that can be attained in the $S^z_{\text{tot}} = 0$ subspace. Time $t$ has been made dimensionless in units of $J_S$ and $\hbar$, i.e. $t' \rightarrow t' J_S / \hbar \equiv t$.

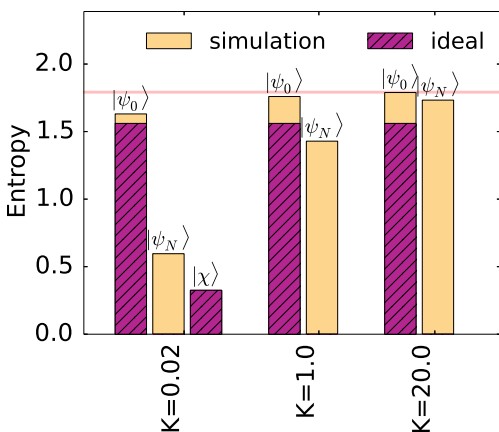

Figure 3: The von Neumann entropy $S(t = 10)$ extracted from Fig. 2 (indicated by simulation) compared to entropy for the mixed-state that corresponds to pure dephasing in the computational basis (marked by ideal). The simulations have been carried out for a central-system of $N_S = 4$ particles strongly coupled ($I = 20 J_S$) to $N_{\mathcal{E}} = 16$ environment particles via random antiferromagnetic Ising coupling. The initial state of the CS and the intra-environment strength (in units of $J_S$) are indicated in the panel. For initial state $|\chi\rangle$ the simulation and ideal entropy exactly coincide and are identical for all three $K$ values ($K = J_S$ and $K = 20 J_S$ have been omitted in the figure). The ideal entropy for state $|\psi_N\rangle$ is zero.

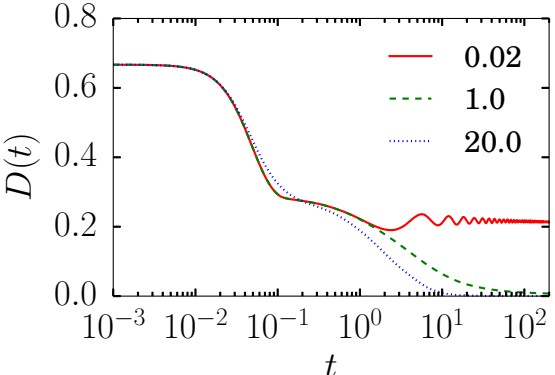

Figure 4: Trace distance [Eq. (19)] between RDMs $\rho_1(0) = |\psi_0\rangle\langle\psi_0|$ and $\rho_2(0) = |\psi_N\rangle\langle\psi_N|$ as a function of dimensionless time $t$. The antiferromagnetic Ising coupling strength is set to $I = 20 J_S$ (strong coupling) and the intra-environment $K$ is indicated in the legend (in units of $J_S$). The system (environment) consists of $N_S = 4$ ($N_{\mathcal{E}} = 16$) particles. Time $t$ has been made dimensionless in units of $J_S$ and $\hbar$, i.e. $t' \rightarrow t' J_S / \hbar \equiv t$. Non-monotonicity is a signature of non-Markovian effects.

## 5.1 Strong interaction

As remarked, the dynamics of the system-environment interaction characterises the only relevant time-scale for the CS in limit 1. That is, the self-Hamiltonian $H_S$ can entirely be neglected in this limit. As a result, the pointer basis is determined by the interaction Hamiltonian, $H_I$. For the present model, Eq. (6), it means that the PS coincide with the spin-up/spin-down basis (henceforth computational basis). We, however, shall study the regime where the system-environment coupling is large, but where $H_S$ can not be neglected. Specifically, the interaction strength is set to $I = 20J_S$ which ensures that the dephasing process due to $H_I$ is short compared to the dynamical time-scale of the antiferromagnet, $H_S$. Moreover, the system consists of $N_S = 4$ and $N_{\mathscr{E}} = 16$ spins. All basis-dependent quantities presented in this subsection, such as the maximum coherence $\mathscr{M}(t)$ and the diagonal components $\rho_{ii}(t)$, refer to the computational basis. Time $t$ is expressed in units of $J_S$ here and in the following.

To reiterate, for any state of the form $|\varphi\rangle = [\alpha\,|\Uparrow\rangle + \beta\,|\Downarrow\rangle]$ the entropy production $\Delta S$ immediately follows from the coefficients. This allows one to tune the desired entropy production by choosing the appropriate coefficients. In anticipation of the simulation results we will study, in addition to initial states $|\psi_0\rangle$ and $|\psi_N\rangle$, the evolution of $|\chi\rangle = [|\Uparrow\rangle + 3\,|\Downarrow\rangle]/\sqrt{10}$.

Consider first the case for which the intra-environment interaction is set to $K = 0.02J_S$. The numerical results of the RDM $\rho(t)$ for $K = 0.02J_S$ are collected in the left column of Fig. 2.

As a result of the strong interaction, the maximum off-diagonal component $\mathscr{M}(t)$ rapidly diminishes for the initial (energy eigen) states $|\psi_0\rangle$ and $|\chi\rangle$, as is seen in the upper row. On the other hand, the Néel-state $|\psi_N\rangle$ being diagonal in $H_I$ initially develops off-diagonal components as a result of the self-Hamiltonian $H_S$ which are subsequently damped in time. Since the environment is rather weak (i.e. small $K$ compared to $I$ and $J_S$), relaxation does not take place and the diagonal components are only slightly perturbed from the initial value, as is illustrated in the middle row. Looking at the bottom row of Fig. 2, it can be seen that the entropy $S(t)$ for $|\psi_N\rangle$ surpasses $1/2$, whilst the maximally attainable value is $S_{\max} \approx 1.79$ in the $S_{\text{tot}}^z = 0$ subspace; this is a considerable amount compared to an ideal zero entropy producing pointer state. And in fact, it develops more entropy than the initial-state $|\chi\rangle$, for which $\Delta S \approx 0.33$. Only in the regime where entropy develops linearly in time does $|\psi_N\rangle$ outperform $|\chi\rangle$ in terms of entropy.

According to the predictability sieve criterion [17], we are thus led to conclude that the state $|\chi\rangle$, which is *not* diagonal in $H_I$, qualifies more as a pointer state than $|\psi_N\rangle$ which *is* diagonal in $H_I$. This is a result of the condition that states which are singled out should not only be minimal entropy producing, but also insensitive to the time which is used for selecting the states [17]. Therefore, it is crucial to go beyond simple perturbative expansions in time. Indeed, our results illustrate how deceptive simple perturbative considerations can be.

One might argue that the robustness of initial-state $|\chi\rangle$ is a pathology of our model Eq. (4), since $|\chi\rangle$ is symmetry protected. In Appendix. B we show that our observations are not restricted to Eq. (4), and holds for more general systems provided that the environment is sufficiently weak.

Let us now discuss the stronger intra-environment interactions $K = J_S$ and $K = 20J_S$. The top row illustrates that $\mathscr{M}(t)$ is slightly more reduced in time compared to $K = 0.02J_S$. By carefully looking at the figure with $K = 20J_S$ near $t = 10^{-1}$ one can notice recohering quantum fluctuations in $\mathscr{M}(t)$ for initial-state $|\chi\rangle$. Calculations in which the number of system-environment connections were varied indicate that the fluctuations are a finite-size effect. The main difference compared to the weak environment ($K = 0.02J_S$) is that the CS now relaxes towards equilibrium in the $S_{\text{tot}}^z = 0$ subspace (for $|\psi_N\rangle$ and $|\psi_0\rangle$). This can be concluded by noting that for $|\psi_N\rangle$ and $|\psi_0\rangle$ the entropy evolves towards $S_{\max}$ of the $S_{\text{tot}}^z = 0$ subspace; or equivalently, the diagonal components grow towards a single value (see centre row Fig. 2), corresponding to the $T = \infty$ configuration.

In Fig. 3 we compare the entropy $S(\tau)$ at $\tau = 10$ with the production which one would expect if the states decohere ideally in the computational basis. That is, the off-diagonal components are entirely quenched and the on-diagonal elements remain unperturbed (pure dephasing). As expected for the state $|\chi\rangle$, the entropy $S(\tau)$ exactly coincides with its ideal value irrespective of $K$. For ground state $|\psi_0\rangle$ the excess entropy relative to the ideal value is rather modest for environment strength $K = 0.02J_S$, in contrast with initial-state $|\psi_N\rangle$. The general picture is that by increasing $K$ one enhances relaxation, which tends towards the maximum entropy as indicated by the red horizontal line.

Note that near the point of relaxation it is no longer possible to speak of preferred states. When the density matrix is proportional to the identity matrix, each basis is on the same footing for it trivially leaves the density matrix diagonal.

Finally, a word on non-Markovian behaviour in our system. In Markovian master equations, the production of thermodynamic entropy is always non-negative provided that a steady state exists [38]. Therefore, the von Neumann entropy oscillations observed in Fig. 2 for $K = 0.02J_S$ point towards non-Markovian behaviour. This can be verified by evaluating the trace distance between two quantum states $\rho_1$ and $\rho_2$

$$D(\rho_1, \rho_2) = \frac{1}{2}\text{Tr}|\rho_1 - \rho_2|, \tag{19}$$

with $|M| = \sqrt{M^\dagger M}$. If $D(\rho_1(t), \rho_2(t))$ is not monotonically decreasing as a function of time $t$ then our system is said to be non-Markovian [39]. In Fig. 4 the trace distance is calculated between the RDMs of the two initial states $|\psi_0\rangle$ and $|\psi_N\rangle$; the three intra-environment strengths are indicated in the legend. The oscillations observed in Fig. 4 for $K = 0.02J_S$ indeed confirm the presence of non-Markovian effects.

## 5.2 Weak interaction

We will now study the decoherence process by a slow environment that is weakly coupled to the CS. The quantum nature of the environment is crucial to have decoherence, as it is based on the development of system-environment entanglement. But from the perspective of the CS, it is helpful to think of the environment as being composed of a large number of magnetic fields which vary slowly in time compared to the systems' intrinsic dynamical scale and shift states of the environment. The adiabatic theorem [40] is applicable in the limit where the level spacing of the CS, $\delta E_S$, is large compared to the dominant frequencies available in the environment. When the energy eigenstates of the CS are non-degenerate, the adiabatic interaction protects these states from measurement [41]. When in addition the interaction is weak, the instantaneous energy eigenstates of the CS closely resemble the initial ($t = 0$) eigenstates [41]. The connection with preferred states was appreciated by Paz and Zurek [4] who identified the energy eigenstates as the pointer basis, given that the environment behaves adiabatically.

In general, the elementary excitations of a quantum system lie very low in energy [42]; for example, in the thermodynamic limit the ground state level spacing, $\delta E_S$, of the antiferromagnetic Heisenberg Hamiltonian goes as $\delta E_S \propto J_S/N_S$ [43]. More specifically, for our system with $N_S = 4$ explicit calculation yields a level spacing of $\delta E_S = J_S$ for all energy levels (apart from degeneracy). To estimate the dominating environmental frequencies, note that individual environmental spins are connected by random isotropic intra-environment strength of order $K$ [see Eq. (8)]. Since there is no temperature suppression of energy levels in $H_{\mathcal{E}}$, the coupling strength $K$ can be used as a crude measure for the intra-environment frequencies.

We shall now examine the range in which it is justified to identify the eigenstates as the pointer basis. To this end, the effect of the environment is studied slightly away from the adiabatic regime. We consider three intra-environment strengths: $K/J_S = 0.02$, 0.2 and 1.

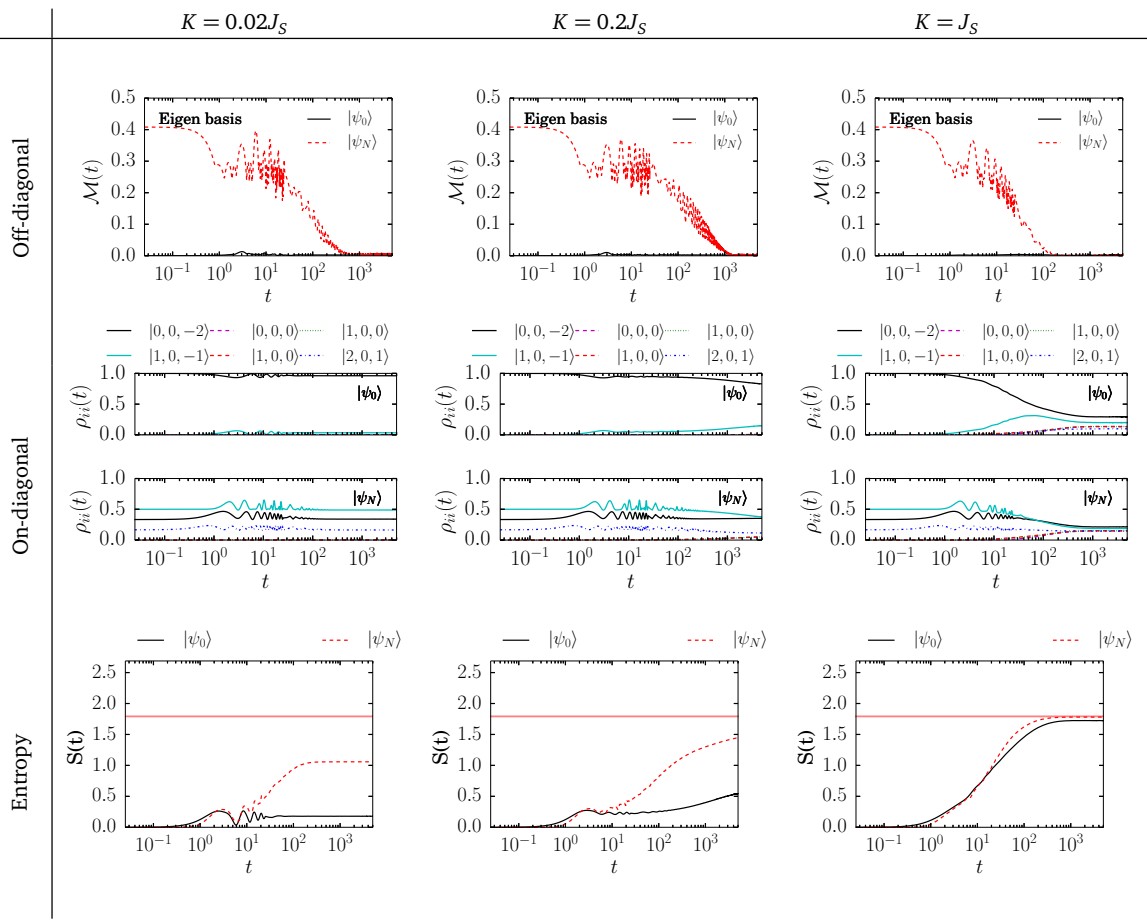

Figure 5: Time-evolution of the reduced density matrix $\rho_{ij}(t)$ in the energy eigenbasis whereby the central-system ($N_S = 4$ particles) is weakly connected ($I = 0.25J_S$) to an environment of $N_{\mathscr{E}} = 16$ particles via random antiferromagnetic Ising coupling. Each column correspond to different intra-environment strengths $K$. The initial states of the central-system (CS) – namely, the ground state $|\psi_0\rangle$ and the Néel-state $|\psi_N\rangle = |\uparrow\downarrow\uparrow\downarrow\rangle$ – are indicated in the panels. The maximum off-diagonal component $\mathscr{M}(t)$ [Eq. (18)] is depicted in the top row. The middle row shows the diagonal components of $\rho(t)$ whereby the states in the legend correspond to quantum numbers $|S_{\text{tot}}, S_{\text{tot}}^z, E\rangle$ where $E$ and $S_{\text{tot}}$ denote the energy eigenvalue and the total spin (of the CS), respectively. The von Neumann entropy $S(t)$ is shown in the bottom row whereby the red horizontal line indicates the maximum value attainable in the $S_{\text{tot}}^z = 0$ subspace. Time $t$ has been made dimensionless in units of $J_S$ and $\hbar$, i.e. $t' \to t'J_S/\hbar \equiv t$.

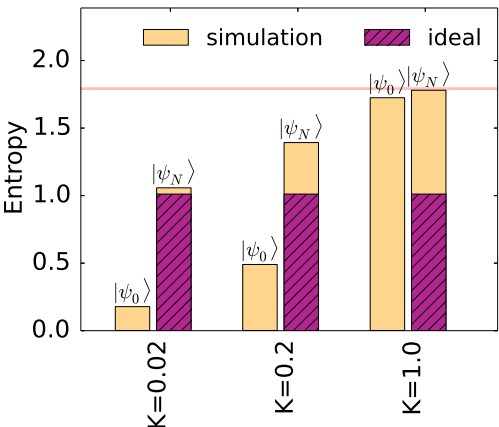

Figure 6: The von Neumann entropy $S(t = 2500)$ extracted from Fig. 5 (indicated by simulation) compared to entropy for the mixed-state that corresponds to pure dephasing in the energy eigenbasis (marked by ideal). The simulations have been carried out for a central-system of $N_S = 4$ particles weakly coupled ($I = 0.25 J_S$) to $N_{\mathcal{E}} = 16$ environment particles via random antiferromagnetic Ising coupling. The initial state of the CS and the intra-environment strength (in units of $J_S$) are indicated in the panel. The ideal entropy for state $|\psi_0\rangle$ is zero.

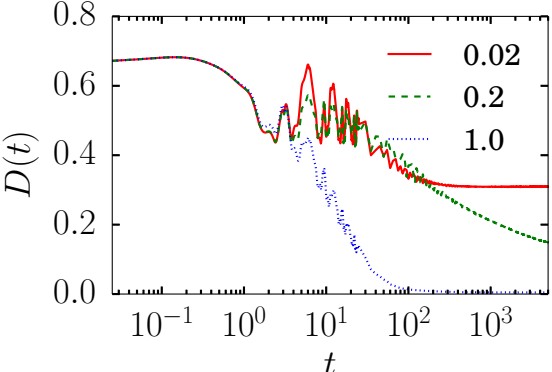

Figure 7: Trace distance [Eq. (19)] between RDMs $\rho_1(0) = |\psi_0\rangle\langle\psi_0|$ and $\rho_2(0) = |\psi_N\rangle\langle\psi_N|$ as a function of dimensionless time $t$. The antiferromagnetic Ising coupling strength is set to $I = 0.25 J_S$ (weak coupling) and the intra-environment $K$ is indicated in the legend (in units of $J_S$). The system (environment) consists of $N_S = 4$ ($N_{\mathcal{E}} = 16$) particles. Time $t$ has been made dimensionless in units of $J_S$ and $\hbar$, i.e. $t' \rightarrow t' J_S/\hbar \equiv t$. Non-monotonicity is a signature of non-Markovian effects.

The interaction strength is set to $I = J_S/4$ and basis-dependent quantities are evaluated in the energy eigenbasis that simultaneously diagonalises $S_{\text{tot}}^z$ and $S_{\text{tot}}^2$, unless specified otherwise. The numerical results pertaining to a $N_S = 4$ RDM coupled to $N_{\mathscr{E}} = 16$ environmental spins are collected in Fig. 5.

We observe that for $K = 0.02$ the environment behaves adiabatically. This can be concluded from the fact that: (i) the ground state $|\psi_0\rangle$ develops no off-diagonal components in the eigenbasis, whilst for the Néel state $|\psi_N\rangle$ the off-diagonal components are exponentially suppressed with time starting from $t \approx 10$ (notwithstanding the oscillations); (ii) the level population of $|\psi_0\rangle$ remains essentially untouched while $|\psi_N\rangle$ is only briefly and slightly perturbed; and (iii) $|\psi_0\rangle$ produces a small amount of entropy. In fact, the amount of entropy that both $|\psi_0\rangle$ and $|\psi_N\rangle$ produce, almost coincides with the entropy production what would be expected if the energy eigenstates are ideal PS. This is depicted in Fig. 6. The excess entropy of both $|\psi_0\rangle$ and $|\psi_N\rangle$ can be further reduced by decreasing $I$, at the expense of performing longer numerical calculations due to the increased decoherence time.

Making the environment stronger by increasing $K$ does not so much affect the loss of coherence, but primarily the characteristic time in which the decoherence occurs (top row Fig. 5). In contrast, the diagonal elements (centre row) and the entropy (bottom row) are significantly perturbed by increasing $K$; a stronger environment enhances relaxation, as was also found in the previous section. This trend is visualised in Fig. 6 whereby the entropy $S(\tau)$ for $\tau = 2500$ is compared with ideal dephasing of the respective initial-state in the eigenbasis. The energy eigenstates are no longer dynamically protected and, as a consequence, the density matrix tends towards (thermodynamic) equilibrium.

The oscillatory behaviour in Fig. 5, all starting near $t \approx 1$, coincides with oscillations in the trace distance $D(t)$ [Eq. (19)] between initial states $|\psi_0\rangle$ and $|\psi_N\rangle$, as shown in Fig. 7. Thus, the transient period in which we find decoherence is marked by strong non-Markovian behaviour, much more so than for a strongly coupled environment (cf. Fig. 4).

## 6 Discussion

As was shown in Sec. 5.1, even when decoherence is extremely fast compared to the typical time-scale of $H_S$, (generic) states that are diagonal in $H_I$ are no longer preferred in terms of predictability. This is to be expected since neglecting $H_S$ amounts to taking the short-time limit when $H_S$ is sufficiently slow. However, in general an explanation of emergent classicality should not be restricted to small time periods only. Indeed, in the frequently discussed quantum Brownian motion example [11], considerations based on time perturbation would (misleadingly) identify spatially localised states as preferred, instead of coherent states. Thus a satisfactory analysis requires that all, perhaps macroscopic, time scales are equally well taken into account.

The results presented in Sec. 5 were performed for a rather modest system of $N_S + N_{\mathscr{E}} = 20$ spins. Whether a small ensemble of spins, as presented here, can convincingly capture all facets of decoherence depends crucially on the environment. For example, a chaotic environment decoheres more efficiently compared to a non-chaotic environment [9]. Furthermore, earlier work has shown that a system with large connectivity within the environment [14, 34, 44] and by employing random couplings [14, 34, 44] an environment of $N_{\mathscr{E}} \approx 16$ spins is sufficiently large to study decoherence [12, 14, 34, 44]. Moreover, in Appendix A it is shown that our considerations are largely unaffected by finite-size effects. Thus no new qualitative features are expected to emerge for much larger environments.

A brief remark on the classical limit of Eq. (5) is now in order. The classical counterpart of Eq. (5) is obtained by replacing spin operators by vectors. As a result, the classical analogue

of the non-degenerate singlet $|\psi_0\rangle$ is the Néel-state $|\psi_N\rangle$. *A priori* one could think that the quantum-to-classical crossover arises from unidirectional coupling to an environment: classicality as a result from the competition between $H_I$, with a well-defined direction in space, and $H_S$, a time-reversal and rotationally invariant self-Hamiltonian. In this work, no evidence was found to substantiate this claim. Instead, $|\psi_N\rangle$ was found to be rather unstable (in terms of entropy) compared to high-energy states $|\Uparrow\rangle$, $|\Downarrow\rangle$, and superpositions thereof.

Clearly, a more realistic model is needed to capture the essential features needed to understand the emergence of classicality in antiferromagnets. For example, the size of the CS presumably plays a role as indicated by experiment [45]. Also the temperature of the environment and the intra-environment interactions might need refinement. These unexplored avenues are left for future work.

## 7 Conclusion

In two mathematical limits the preferred basis problem seems solved, namely: (a) the strong system-environment coupling limit and (b) the weak-coupling and slow-environment limit [4]. The preferred states corresponding to limit (a) [limit (b)] are diagonal in the interaction Hamiltonian $H_I$ [self-Hamiltonian $H_S$].

In this work, a more physical line of approach was adopted, whereby decoherence *near* the two limits was considered. Specifically, the decoherence of a small antiferromagnetic system has been (numerically) considered with special emphasis on pointer states (PS). Both the loss of coherence and the predictability (or robustness) of PS was stressed. Statistical entropy was used as a means to quantify the robustness of specific states.

Two initial states of the central-system were considered in particular – namely $|\psi_0\rangle$, the ground state of $H_S$, and the Néel-state $|\psi_N\rangle = |\uparrow\downarrow\uparrow\downarrow\rangle$. For these states it was found that near limit (a) [limit (b)] the coherences of the reduced density matrix $\rho(t)$ evaluated in the $H_I$ [$H_S$] basis are quenched, irrespective of the strength of the environment self-Hamiltonian $H_{\mathscr{E}}$. However, the dynamical timescale of $H_{\mathscr{E}}$ determines to a large extent the robustness of specific states, since stronger environments enhance relaxation within the $S_{\text{tot}}^z$ subspaces. Hence, our work suggests that dynamically fast environments tend to erase preferences for a specific basis.

Conversely, when $H_{\mathscr{E}}$ is sufficiently slow it was found that near limit (b), indeed, the energy eigenstate $|\psi_0\rangle$ appears robust against environmental coupling. But importantly, near limit (a) we demonstrated by explicit example, that specific states *not* diagonal in $H_I$ can become more stable (i.e., more PS-like) than typical states that *are* diagonal in $H_I$ (such as $|\psi_N\rangle$). Thus, neglecting $H_S$ seems justified in e.g. a quantum measurement set-up where the interaction is subsequently turned off [25]. On macroscopic time scales, however, care must be taken in analysing PS since a (perturbatively) small self-Hamiltonian $H_S$ can become dynamically relevant.

## Acknowledgements

MIK and HCD acknowledges financial support by the European Research Council, project 338957 FEMTO/NANO. We thank Edo van Veen for helpful discussions and proof reading.

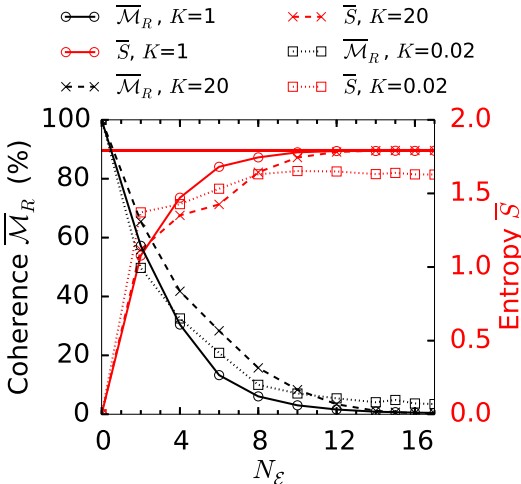

(a) Strong coupling $I = 20J_S$ and initial-state $|\psi_0\rangle$; coherence suppression $\overline{\mathcal{M}}_R(180, 20)$ is evaluated in the computational basis.

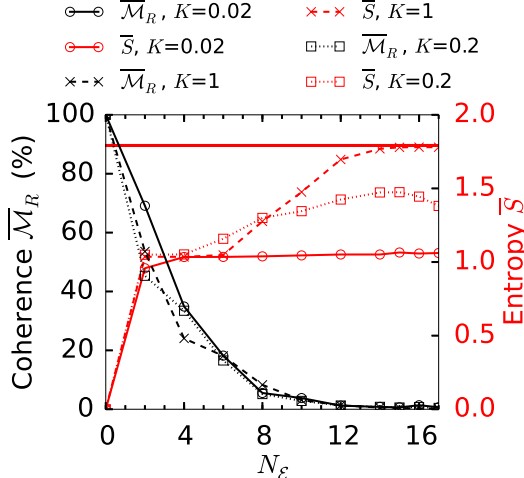

(b) Weak coupling $I = 0.25J_S$ and initial-state $|\psi_N\rangle$; coherence suppression $\overline{\mathcal{M}}_R(4500, 500)$ is evaluated in the energy eigen basis.

Figure 8: The time-averaged entropy and maximum off-diagonal component as a function of the environment size $N_{\mathscr{E}}$. The intra-environment strength is fixed according to $KN_{\mathscr{E}}(N_{\mathscr{E}} - 1) = \text{const}$; the various values of $K$ indicated in the panels correspond to $N_{\mathscr{E}} = 16$.

## A Finite-size effects

To convince the reader that the present results do not significantly suffer from finite-size artifacts, we have varied the number of environment spins $N_{\mathscr{E}}$. In order to make a fair comparison between environments of different size, the intra-environment strength $KN_{\mathscr{E}}(N_{\mathscr{E}} - 1)$ is held fixed since the number of elements $K_{ab} \neq 0$ goes as $\propto N_{\mathscr{E}}(N_{\mathscr{E}} - 1)$. In addition, time-averaging is performed on the entropy $S$ and the coherence suppression $\mathcal{M}$, i.e.

$$\overline{\mathcal{M}}(t_\infty, \Delta T) \equiv \frac{1}{\Delta T} \int_{t_\infty}^{t_\infty + \Delta T} \mathcal{M}(t) \, dt \, . \tag{20}$$

Time-averaging ensures that our conclusions are insensitive to small variations in time, as the fast oscillations in small-$N_{\mathscr{E}}$ environments are averaged out. The results for the strong interaction regime (as discussed in Sec. 5.1) with $I = 20J_S$ are presented in Fig. 8a. The initial-state is set to $|\psi_0\rangle$ and $\overline{\mathcal{M}}_R(t_\infty, \Delta T) = 100 \cdot \overline{\mathcal{M}}(t_\infty, \Delta T)/\mathcal{M}(0)$ is calculated in the computational basis. The exponential decrease in $\overline{\mathcal{M}}_R$ observed for $K = J_S$ and $K = 20J_S$ are well captured by the simple $\overline{\mathcal{M}}_R \propto 2^{-N_{\mathscr{E}}/2}$ scaling law [44]. Both the entropy production, $\overline{S}$, and the coherence suppression, $\overline{\mathcal{M}}_R$, level off around $N_{\mathscr{E}} = 10$. Thus an environment of $N_{\mathscr{E}} = 10$ spins is already sufficiently large to capture the main features.

In Fig. 8b we collect the results for interaction strength $I = J_S/4$ with initial-state $|\psi_N\rangle$. Note that $\overline{\mathcal{M}}_R$ is now calculated in the eigenbasis. The entropy of the slow environment, marked by $K = 0.02$ in the panel, is insensitive to the size of the environment for $N_{\mathscr{E}} \geq 4$. Although the environment size $N_{\mathscr{E}} = 16$ is insufficiently large to rule out finite-size effects for the two other environment strengths, it appears that the increase of entropy as a function of $N_{\mathscr{E}}$ levels off around $N_{\mathscr{E}} = 14$. The same scaling behaviour of $\overline{\mathcal{M}}_R$ as a function of $N_{\mathscr{E}}$ was found as in Fig. 8a, but now in the basis that diagonalises $H_S$.

# B   Symmetry broken central-system Hamiltonian

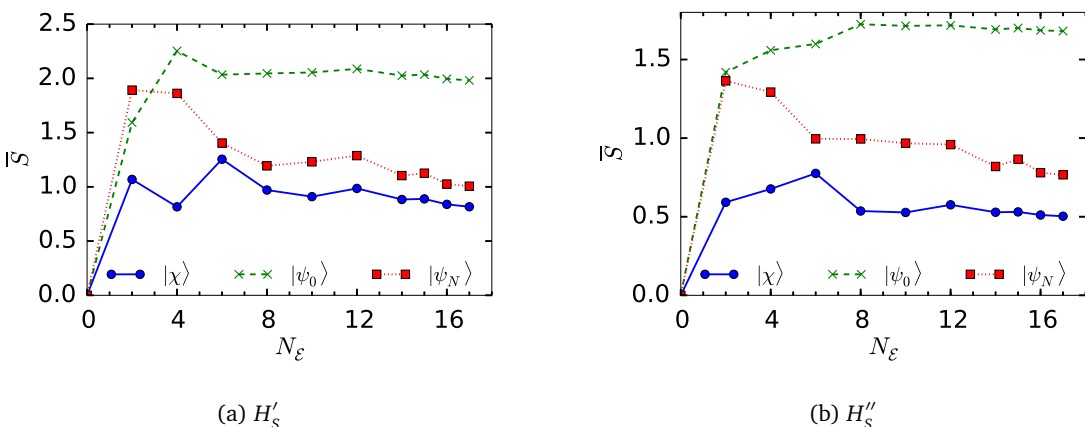

(a) $H'_S$                                              (b) $H''_S$

Figure 9: Effect of an $S^z_{\text{tot}}$ symmetry breaking (additional) term (indicated below the panel) on the time-averaged entropy $\overline{S}(180, 20)$, as a function of the environment size $N_{\mathcal{E}}$. The CS is strongly coupled to the environment with strength $I = 20J_S$, while the intra-environment strength is fixed to $KN_{\mathcal{E}}(N_{\mathcal{E}} - 1) = 24/5\,J_S$. The dipolar coupling $J^{xx}_S$ and transverse field $h^x$ are both set to $J_S$.

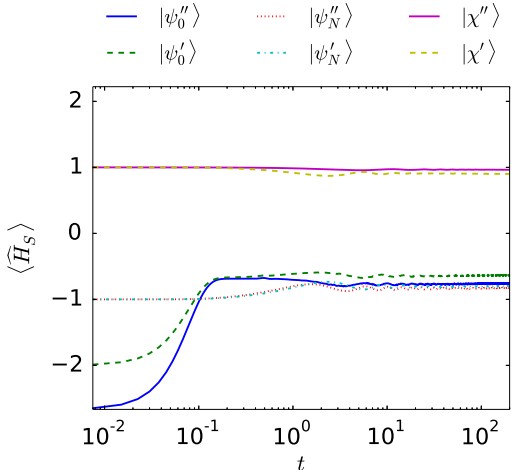

Figure 10: Energy of the CS (in units of $J_S$) for a Hamiltonian that contains an anisotropy $\widehat{H}_S = H_S + H'_S$ or $\widehat{H}_S = H_S + H''_S$. The primes on the initial states in the legend refer to the Hamiltonian with the respective anisotropy. The environment consists of $N_{\mathcal{E}} = 16$ particles with coupling strength $K = 0.02J_S$. Time $t$ has been made dimensionless in units of $J_S$ and $\hbar$, i.e. $t' \rightarrow t'J_S/\hbar \equiv t$.

As a result of Hamiltonians $H_S$ and $H_I$ (Eqs. (5) and (6), respectively) the initial state $|\chi\rangle$ is protected from relaxation by the $S^z_{\text{tot}}$ quantum number. It will now be shown that the observations in Sec. 5 are not restricted to symmetry protected states. Accordingly, a symmetry breaking term is added to $H_S$ such that $S^z_{\text{tot}}$ is no longer conserved. Two such terms are considered: a transverse field $H'_S = h^x S^x_{\text{tot}}$ and $H''_S = J^{xx}_S \sum_{\langle i,j \rangle} S^x_i S^x_j$ that mimicks a near-neighbour dipole-dipole interaction [46] $\propto \sum_{\langle i,j \rangle} [\mathbf{S}_i \cdot \mathbf{S}_j - 3(\mathbf{n} \cdot \mathbf{S}_i)(\mathbf{n} \cdot \mathbf{S}_j)]$ where $\mathbf{n} \parallel \hat{x}$. The new self-Hamiltonian is thus $\widehat{H}_S = H_S + H'_S$ or $\widehat{H}_S = H_S + H''_S$. The (time-averaged) entropy $\overline{S}(t_\infty, \Delta T)$

[see Eq. (20)] of the CS with additional anisotropy $H'_S$ or $H''_S$ are indicated in Fig. 9. What is seen is that for all sizes $N_{\mathscr{E}}$ the entropy of initial state $|\chi\rangle$ is lower than that of $|\psi_0\rangle$ and $|\psi_N\rangle$. Thus, $|\chi\rangle$ is more robust compared to the other two states, while it is no longer symmetry protected from relaxation. One possible explanation is that $|\chi\rangle$ is still relatively high in energy compared to the mean energy of the CS ($2^{-N_S}\sum_{i=1}^{2^{N_S}}\widehat{E}^i_S = 0$ in both cases) and is, as a result of the low energy content of the $T = \infty$ environment, unable to lose some of the CS energy to *excite* lower lying energy states. Or put differently, the entropy production for the state $|\chi\rangle$ is energetically suppressed. In Fig. 10 the time-development of the energy is depicted for the different systems. Indeed, what is seen is that over the course of time the energy difference for $|\psi_N\rangle$ grows towards $\Delta E = 0.20$ and $\Delta E = 0.17$ with respectively $H'_S$ and $H''_S$. In comparison, the energy difference for $|\chi\rangle$ is found to be much smaller: $\Delta E = -0.10$ and $\Delta E = -0.04$ with $H'_S$ and $H''_S$, respectively. Moreover, numerical calculations (not shown here) indicate that initial-state $|\chi\rangle$ readily overtakes $|\psi_0\rangle$ and $|\psi_N\rangle$ in entropy when increasing $K$.

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
