# Peer review of "Decoherence and pointer states in small antiferromagnets: A benchmark test"

_SciPost Physics, doi:SciPost Phys. 2, 010 (2017)_

## Round 1 · Referee Report · Anonymous · 2016-12-29

Strengths
-The authors address here a rather fundamental issue: namely finding the pointer states, the most immune states with respect to environmental decoherence, of an interacting system, here a small collection of coupled spin ½.
- The results look sound and reliable.
- The main result of the paper concerns the strong system-environemnt regime. The author show explicitly that contrary to the naive expectation, some stateswhich are not diagonal in the interaction Hamiltonian $H_I$ can become more stable with time than typical states that are diagonal in $H_I$.
Weaknesses
- The authors studies only very small systems (4 spins 1/2 for the system coupled to 16 spins 1/2 for the environment). The system is far from an experimentally realistic system as studied in [42].
- The emergence of classicality cannot be seriously studied with such small systems.
- As such, this study is more of an exercise to illustrate that point states should not be chosen with care especially in the strong coupling case.
- Figures'caption should be more self-contained. We are always obliged to dig in the text the values of the chosen parameters.
Report
I am somewhat mixed by this paper. As I wrote above, this is more of a simple numerical exercise to look at the dynamics of a very small Heisenberg AF system (4 spins) coupled to a small environment of 16 spin 1/2. In that sense, this is not very original nor very conclusive for more realistic systems.
However, this study has the merit to simply illustrate that pointer states are not always the most intuitive ones particularly in the strong system-environment coupling where one would think at first sight that states diagonal in $H_I$ would be appropriate pointer states.
Requested changes
- Please include the values of the parameters studied in the simulation in the captions of Fig 2-5. This would ease the reading of the text.
- A few words commenting Fig 3& Fig 5 would be welcome in the caption (or in the text): why $|\chi>$ has disappeared for $K=1$ and $K=20$ ? When in the ideal case, the entropy is zero, please write it.

---

## Round 1 · Referee Report · Anonymous · 2017-1-4

Strengths
1) Numerical analysis of a very important problem, the determination of the pointer states.
2) The analysis of the strong coupling limit suggests non-trivial effects
Weaknesses
1) The analysis is restricted to few cases a discussion on how to draw general statements from the simulations is limited
2) The presentation can be considerably improved
Report
The paper by Donker et al deals with a very important problem, the identification of the pointer states for a quantum system coupled to an external bath. Through a numerical integration of the equation of motion they face the question, both in the strong and weak coupling limit, on how to identify the pointer states. As already said, the question is important and the paper hints at clarifying some interesting aspects of the issue.
The model analysed by the authors consists of a four-site antiferromagnetic coupled through a z-z coupling to a spin bath. Both the system-bath and the intra-bath couplings are random in magnitude. Because of numerical limitations (as far as I understand) the analysis is limited to few initial configurations.
Starting from the two extreme limits in the strong and weak coupling regimes the authors develops the numerics around those showing that some non-trivial effects appears. Personally I found the results in the strong coupling regime of particular interest.
Despite this positive judgment I find that there are several other aspects of the paper that need to be improved. If not in additional simulations, I think that a revised presentation will certainly help. I will list below the points where the paper would benefit from some additional discussion
- I am slightly confused from the introduction of the choice of the initial states. If on one side I see the reason for this choice, on the other side it seems implied that there are sizeable non-Markovian effects. It seems to me that the different degree on this "non-Markovian behaviour" and its differences going from the weak to the strong coupling are not addressed. I guess a discussion of this point is relevant for the analysis (or at least the authors should point out why/if it is not relevant)
- I was unable to find in the paper more informations on the choice of the random couplings. It seemed in reading that a single choice of the couplings was presented
- I miss to see the reason of choosing 4-sites for the system instead of 2 (for example). Is there something that we learn from this choice?
- The presentation may also improve considerably in the introduction and definition of various quantities. For example: the reduced density matrix \rho is introduced in page 2 and defined in page 5, similarly for the Hamiltonian H
- I think that the discussion on how to draw some general conclusions from the analysis of few cases can be improved.
- The format of Fig 2 and 4 should be improved, some features discussed in the text are difficult to visualise.
Requested changes
These are listed in the report

---

## Round 2 · Referee Report · Anonymous (Referee 1) · 2017-2-14

Strengths
Weaknesses
Report
They have improved the presentation of their results using a log scale for the time evolution, added a few precision here and there notably concerning the discussion of finite size effects.
I therefore recommend publication of this manuscript.
Requested changes
Could you add somewhere in the text or in the captions the unit of the time evolution t (\hbar/J_s) ?

---

## Round 2 · Referee Report · Anonymous (Referee 2) · 2017-2-15

Strengths
Weaknesses
Report
Requested changes
none

---

## Round 2 · Author Response

Below, we comment on the remarks made in the reports.

---

## Round 2 · List of Changes

Warnings issued while processing user-supplied markup:
- Inconsistency: Markdown and reStructuredText syntaxes are mixed. Markdown will be used.
Add "#coerce:reST" or "#coerce:plain" as the first line of your text to force reStructuredText or no markup.
You may also contact the helpdesk if the formatting is incorrect and you are unable to edit your text.
Report 60
(1) I am slightly confused from the introduction of the choice of the initial states. If on one side I see the reason for this choice, on the other side it seems implied that there are sizeable non-Markovian effects. It seems to me that the different degree on this "non-Markovian behaviour" and its differences going from the weak to the strong coupling are not addressed. I guess a discussion of this point is relevant for the analysis (or at least the authors should point out why/if it is not relevant)
Reply: * We agree that this is an important aspect not covered in the work. To address this point we have calculated the trace distance between the reduced density matrices with initial states and |psi_0> and |psi_N> (the non-monotonicity of this quantity expresses non-Markovian behaviour), for both strong and weak coupling (see Figs. 4 & 7). A paragraph, discussing Figs. 4 and 7, is added at the bottom of Secs. 5.1 and 5.2, respectively.
(2) - I was unable to find in the paper more informations on the choice of the random couplings. It seemed in reading that a single choice of the couplings was presented
Reply: * Our paper was indeed not very explicit about the precise nature of the random couplings. An additional paragraph was added in Sec. 3.3 to correct for this.
(3) - I miss to see the reason of choosing 4-sites for the system instead of 2 (for example). Is there something that we learn from this choice?
Reply: * The 2-site central-system is rather constrained due to its limited size. In the S^z_tot=0 subspace, the CS can in fact be mapped onto a two-level system. Therefore, we choose to consider the 4-site central system. We briefly discuss this point at the bottom paragraph of Sec. 3.2.
(4) The presentation may also improve considerably in the introduction and definition of various quantities. For example: the reduced density matrix \rho is introduced in page 2 and defined in page 5, similarly for the Hamiltonian H
Reply: * These suggestions have been implemented.
(5) I think that the discussion on how to draw some general conclusions from the analysis of few cases can be improved.
Reply: * We agree that certain sentences can be interpreted to general if one were to only read the Conclusion. We have revised part of the Conclusion by using a more careful choice of words.
(6) The format of Fig 2 and 4 should be improved, some features discussed in the text are difficult to visualise.
Reply: * To improve readability we have put time t in Figs. 2 & 4 on logaritmic scale and increased the font. In addition, extra simulations have been carried out such that multiple orders of magnitude in time are visible.
== Report 58
(7) The authors studies only very small systems (4 spins 1/2 for the system coupled to 16 spins 1/2 for the environment). The system is far from an experimentally realistic system as studied in [42]. - The emergence of classicality cannot be seriously studied with such small systems.
Reply: * The fact that 16 environment spins is enough to study decoherence processes is indeed counter intuitive, and was not discussed in our work. In the Discussion (Sec. 6) we now stress the importance of the precise details of the environment and refer to the appropriate references.
(8) Please include the values of the parameters studied in the simulation in the captions of Fig 2-5. This would ease the reading of the text. (9) A few words commenting Fig 3& Fig 5 would be welcome in the caption (or in the text): why |χ> has disappeared for K=1 and K=20 ? When in the ideal case, the entropy is zero, please write it.
Reply: * The requested changes have been implemented.

---

## Round 3 · List of Changes

1) The figures have been enlarged, in particular Figs. 2 & 5.
2) The units of time t is stressed in the captions of the figures.

---

## Editorial Decision

published